# Statistical Model Aggregation via Parameter Matching

**Mikhail Yurochkin**[1,2]
mikhail.yurochkin@ibm.com

**Mayank Agarwal**[1,2]
mayank.agarwal@ibm.com

**Soumya Ghosh**[1,2,3]
ghoshso@us.ibm.com

**Kristjan Greenewald**[1,2]
kristjan.h.greenewald@ibm.com

**Trong Nghia Hoang**[1,2]
nghiaht@ibm.com

IBM Research,[1] MIT-IBM Watson AI Lab,[2] Center for Computational Health[3].

## Abstract

We consider the problem of aggregating models learned from sequestered, possibly heterogeneous datasets. Exploiting tools from Bayesian nonparametrics, we develop a general meta-modeling framework that learns shared global latent structures by identifying correspondences among local model parameterizations. Our proposed framework is model-independent and is applicable to a wide range of model types. After verifying our approach on simulated data, we demonstrate its utility in aggregating Gaussian topic models, hierarchical Dirichlet process based hidden Markov models, and sparse Gaussian processes with applications spanning text summarization, motion capture analysis, and temperature forecasting.[1]

## 1 Introduction

One is often interested in learning from groups of heterogeneous data produced by related, but unique, generative processes. For instance, consider the problem of discovering shared topics from a collection of documents, or extracting common patterns from physiological signals of a cohort of patients. Learning such shared representations can be relevant to many heterogeneous, federated, and transfer learning tasks. Hierarchical Bayesian models [3, 12, 29] are widely used for performing such analyses, as they are able to both naturally model heterogeneity in data and share statistical strength across heterogeneous groups.

However, when the data is large and scattered across disparate silos, as is increasingly the case in many real-world applications, use of standard hierarchical Bayesian machinery becomes fraught with difficulties. In addition to costs associated with moving large volumes of data, the computational cost of full Bayesian inference may be prohibitive. Moreover, pooling sequestered data may also be undesirable owing to concerns such as privacy [11]. While distributed variants [18] have been developed, they require frequent communication with a central server and hence are restricted to situations where sufficient communication bandwidth is available. Yet others [26] have proposed federated learning algorithms to deal with such scenarios. However, these algorithms tend to be bespoke and can require significant modifications based on the models being federated.

Motivated by these challenges, in this paper we develop Bayesian nonparametric *meta-models* that are able to coherently combine models trained on independent partitions of data (model fusion). Relying on tools from Bayesian nonparametrics (BNP), our meta model treats the parameters of the locally trained models as noisy realizations of latent global parameters, of which there can be

infinitely many. The generative process is formally characterized through a Beta-Bernoulli process (BBP) [31]. Model fusion, rather than being an ad-hoc procedure, then reduces to posterior inference over the meta-model. Governed by the BBP posterior, the meta-model allows local parameters to either match existing global parameters or create *new* ones. This ability to grow or shrink the number of parameters is crucial for combining local models of varying complexity – for instance, hidden Markov models with differing numbers of states.

Our construction provides several key advantages over alternatives in terms of scalability and flexibility. First, scaling to large data through parallelization is trivially easy in our framework. One would simply train the local models in parallel and fuse them. Armed with a Hungarian algorithm-based efficient MAP inference procedure for the BBP model, we find that our train-in-parallel and fuse scheme affords significant speedups. Since our model fusion procedure is independent of the learning and inference algorithms that may have been used to train individual models, we can seamlessly combine models trained using disparate algorithms. Furthermore, since we only require access to trained local models and not the original data, our framework is also applicable in cases where only pre-trained models are available but not the actual data, a setting that is difficult for existing federated or distributed learning algorithms.

Finally, we note that our development is largely agnostic to the form of the local models and is reusable across a wide variety of domains. In fact, up to the choice of an appropriate base measure to describe the local parameters, the exact same algorithm can be used for fusion across qualitatively different settings. We illustrate this flexibility by demonstrating proficiency at combining a diverse class of models, which include sparse Gaussian processes, mixture models, topic models and hierarchical Dirichlet process based hidden Markov models.

## 2   Background and Related Work

Here, we briefly review the building blocks of our approach and highlight the differences of our approach from existing work.

**Indian Buffet Process and the Beta Bernoulli Process**   The Indian buffet process (IBP) specifies a distribution over sparse binary matrices with infinitely many columns [17]. It is commonly described through the following culinary metaphor. Imagine $J$ customers arrive sequentially at a buffet and choose dishes to sample. The first customer to arrive samples Poisson($\gamma_0$) dishes. The $j$-th subsequent customer then tries each of the dishes selected by previous customers with probability proportional to the dish's popularity, and then additionally samples Poisson($\gamma_0/j$) new dishes that have not yet been sampled by any customer. Thibaux and Jordan [31] showed that the de Finetti mixing distribution underlying the IBP is a Beta Bernoulli Process (BBP). Let $Q$ be a random measure drawn from a Beta process, $Q \mid \alpha, \gamma_0, H \sim \mathrm{BP}(\alpha, \gamma_0 H)$, with mass parameter $\gamma_0$, base measure $H$ over $\Omega$ such that $H(\Omega) = 1$ and concentration parameter $\alpha$. It can be shown $Q$ is a discrete measure $Q = \sum_i \mathrm{q}_i \delta_{\theta_i}$ formed by an infinitely countable set of (weight, atom) pairs $(\mathrm{q}_i, \theta_i) \in [0,1] \times \Omega$. The weights $\{\mathrm{q}_i\}_{i=1}^{\infty}$ are distributed by a stick-breaking process [30], $\nu_1 \sim \mathrm{Beta}(\gamma_0, 1)$, $\nu_i = \prod_{j=1}^{i} \nu_j$ and the atoms $\theta_i$ are drawn i.i.d. from $H$. Subsets of atoms in $Q$ are then selected via a Bernoulli process. That is, each subset $\mathcal{T}_j$ with $j = 1, \ldots, J$ is characterized by a Bernoulli process with base measure $Q$, $\mathcal{T}_j \mid Q \sim \mathrm{BeP}(Q)$. Consequently, subset $\mathcal{T}_j$ is also a discrete measure formed by pairs $(\mathrm{b}_{ji}, \theta_i) \in \{0,1\} \times \Omega$, $\mathcal{T}_j := \sum_i \mathrm{b}_{ji} \delta_{\theta_i}$, where $\mathrm{b}_{ji} \mid \mathrm{q}_i \sim \mathrm{Bernoulli}(\mathrm{q}_i) \, \forall i$ is a binary random variable indicating whether atom $\theta_i$ belongs to subset $\mathcal{T}_j$. The collection of such subsets is then said to be distributed by a Beta-Bernoulli process. Marginalizing over the Beta Process distributed $Q$ we recover the predictive distribution, $\mathcal{T}_J \mid \mathcal{T}_1, \ldots, \mathcal{T}_{J-1} \sim \mathrm{BeP}\left(\frac{\alpha \gamma_0}{J + \alpha - 1} H + \sum_i \frac{m_i}{J + \alpha - 1} \delta_{\theta_i}\right)$, where $m_i = \sum_{j=1}^{J-1} \mathrm{b}_{ji}$ (dependency on $J$ is suppressed for notational simplicity) which can be shown to be equivalent to the IBP. Our work is related to recent advances [33] in efficient BBP MAP inference.

**Distributed, Decentralized, and Federated Learning**   Similarly to us, federated and distributed learning approaches also attempt to learn from sequestered data. These approaches roughly fall into two groups, those [9, 15, 21, 22, 23] that decompose a global, centralized learning objective into localized ones that can be optimized separately using local data, and those that iterate between training local models on private data sources and distilling them into a global model [8, 6, 18, 26]. The former group carefully exploits properties of the local models being combined. It is unclear how

methods developed for a particular class of local models (for example, Gaussian processes) can be adapted to a different class of models (say, hidden Markov models). More recently, [32] also exploited a BBP construction for federated learning, but were restricted to only neural networks. Alternatively, [20] follows a different development that requires local models of different classes to be distilled into the same class of surrogate models before aggregating them, which, however, accumulates local distillation error (especially when the number of local models is large). Members of the latter group require frequent communication with a central server, are poorly suited to bandwidth limited cases, and are not applicable when the pretrained models cannot share their associated data. Others [5] have proposed decentralized approximate Bayesian algorithms. However, unlike us, they assume that each of the local models have the same number of parameters, which is unsuitable for federating models with different complexities.

## 3 Bayesian Nonparametric Meta Model

We propose a Bayesian nonparametric meta model based on the Beta-Bernoulli process [31]. In seeking a "meta model", our goal will be to describe a model that generates collections of parameters that describe the local models. This meta model can then be used to infer the parameters of a global model from a set of local models learned independently on private datasets.

Our key assumption is that there is an unknown shared set of parameters of unknown size across datasets, which we call global parameters, and we are able to learn subsets of noisy realizations of these parameters from each of the datasets, which we call local parameters. The noise in local parameters is motivated by estimation error due to finite sample size and by variations in the distributions of each of the datasets. Additionally, local parameters are allowed to be permutation invariant, which is the case in a variety of widely used models (e.g., any mixture or an HMM).

We start with Beta process prior on the collection of global parameters, $G \sim \mathrm{BP}(\alpha, \gamma_0 H)$ then $G = \sum_i p_i \delta_{\theta_i}$, $\theta_i \sim H$, where $H$ is a base measure, $\theta_i$ are the global parameters, and $p_i$ are the stick breaking weights. To devise a meta-model applicable to broad range of existing models, we do not assume any specific base measure and instead proceed with general exponential family base measure,

$$p_\theta(\theta \mid \tau, n_0) = \mathcal{H}(\tau, n_0) \exp(\tau^T \theta - n_0 \mathcal{A}(\theta)). \tag{1}$$

Local models do not necessarily have to use all global parameters, e.g. a Hidden Markov Model for a given time series may only contain a subset of latent dynamic behaviors observed across collection of time series. We use a Bernoulli process to allow $J$ local models to select a subset of global parameters,

$$Q_j \mid G \sim \mathrm{BeP}(G) \text{ for } j = 1, \ldots, J. \tag{2}$$

Then $Q_j = \sum_i b_{ji} \delta_{\theta_i}$, where $b_{ji} \mid p_i \sim \mathrm{Bern}(p_i)$ is a random measure representing the subset of global parameters characterizing model $j$. We denote the corresponding subset of indices of the global parameters induced by $Q_j$ as $\mathcal{C}_j = \{i : b_{ji} = 1\}$. The noisy, permutation invariant local parameters estimated from dataset $j$ are modeled as,

$$v_{jl} \mid \theta_{c(j,l)} \sim F(\cdot \mid \theta_{c(j,l)}) \text{ for } l = 1, \ldots, L_j, \tag{3}$$

where $L_j = \mathrm{card}(\mathcal{C}_j)$ and $c(j,l) : \{1, \ldots, L_J\} \to \mathcal{C}_j$ is an unknown mapping of indices of local parameters to indices of global parameters corresponding to dataset $j$. Parameters of different models of potential interest may have different domain spaces and domain-specific structure. To preserve generality of our meta-modeling approach we again consider a general exponential family density for the local parameters,

$$p_v(v \mid \theta) = h(v) \exp(\theta^T T(v) - \mathcal{A}(\theta)). \tag{4}$$

where $T(\cdot)$ is the sufficient statistic function.

**Interpreting the model.** We emphasize that our construction describes a *meta model*, in particular it describes a generative process for the parameters of the local models rather than the data itself. These parameters are "observed" either when pre-trained local models are made available or when the local models are learned independently and potentially in parallel across datasets. The meta model then infers shared latent structure among the datasets. The Beta process concentration parameter $\alpha$ controls the degree of sharing across local models while the mass parameter $\gamma_0$ controls the number of global parameters. The interpretation of the exponential family parameters, $\tau$ and $n_0$, depends on

the choice of the particular exponential family. We provide a concrete example with the Gaussian distribution in Section 4.1.

Several prior works [10, 19, 4] explore the meta modeling perspective. The key difference with our approach is that we consider broader model class allowing for inherent *permutation invariant* structure of the parameter space, e.g. mixture models, topic models, hidden Markov models and sparse Gaussian processes. The aforementioned approaches are only applicable to models with natural parameter ordering, e.g. linear regression, which is a simpler special case of our construction. Permutation invariance leads to inferential challenges associated with finding correspondences across sets of local parameters and learning the size of the global model, which we address in the next section.

## 4 Efficient Meta Model Inference

Taking the optimization perspective, our goal is to maximize the posterior probability of the global parameters given the local ones. Before discussing the objective function we re-parametrize (4) to side-step the index mappings $c(\cdot, \cdot)$ as follows:

$$v_{jl} \mid B, \theta \sim F\left(\cdot \mid \sum_i B_{il}^j \theta_i\right) \text{ s.t. } \sum_i B_{il}^j = 1, b_{ji} = \sum_l B_{il}^j \in \{0, 1\}, \tag{5}$$

where $B = \{B_{il}^j\}_{i,j,l}$ are the assignment variables such that $B_{il}^j = 1$ denotes that $v_{jl}$ is matched to $\theta_i$, i.e. $v_{jl}$ is the local parameter realization of the global parameter $\theta_i$; $B_{il}^j = 0$ implies the opposite.

The objective function is then $\mathbb{P}(\theta, B \mid v, \Theta)$, where $\Theta = \{\tau, n_0\}$ are the hyperparameters and indexing is suppressed for simplicity. In the context of our meta model this problem has been studied when distributions in (1) and (4) are Gaussian [32] or von Mises-Fisher [33], which are both special cases of our meta model. However, this objective requires $\Theta$ to be chosen a priori leading to potentially sub-optimal solutions or to be selected via expensive cross-validation.

We show that it is possible to simplify the optimization problem via integrating out $\theta$ and jointly learn hyperparameters and matching variables $B$, all while maintaining the generality of our meta model. Define $Z_i = \{(j, l) \mid B_{il}^j = 1\}$ to be the index set of the local parameters assigned to the $i$th global parameter, then the objective functions we consider is,

$$\mathcal{L}(B, \Theta) = \mathbb{P}(B \mid v) \propto \mathbb{P}(B) \int p_v(v \mid B, \theta) p_\theta(\theta) \, \mathrm{d}\theta = \mathbb{P}(B) \prod_i \int \prod_{z \in Z_i} p_v(v_z \mid \theta_i) p_\theta(\theta_i) \, \mathrm{d}\theta_i$$

$$= \mathbb{P}(B) \prod_i \mathcal{H}(\tau, n_0) \int \left(\prod_{z \in Z_i} h(v_z)\right) \exp\left((\tau + \sum_{z \in Z_i} T(v_z))^T \theta_i - (\mathrm{card}(Z_i) + n_0)\mathcal{A}(\theta)\right) \, \mathrm{d}\theta_i \tag{6}$$

$$= \mathbb{P}(B) \prod_i \frac{\mathcal{H}(\tau, n_0) \prod_{z \in Z_i} h(v_z)}{\mathcal{H}(\tau + \sum_{z \in Z_i} T(v_z), \mathrm{card}(Z_i) + n_0)},$$

Holding $\Theta$ fixed, and then taking the logarithm and noting that $\sum_i \sum_{j,l} B_{il}^j \log h(v_{jl})$ is constant in $B$, we wish to maximize,

$$\mathcal{L}_\Theta(B) = \log \mathbb{P}(B) - \sum_i \log \mathcal{H}\left(\tau + \sum_{j,l} B_{il}^j T(v_{jl}), \sum_{j,l} B_{il}^j + n_0\right), \tag{7}$$

where we have used $\mathcal{L}_\Theta(B)$ to denote the objective when $\Theta$ is held constant. Despite the large number of discrete variables, we show that $\mathcal{L}_\Theta(B)$ admits a reformulation that permits efficient inference by iteratively solving small sized linear sum assignment problems (e.g., the Hungarian algorithm [25]).

We consider iterative optimization where we optimize the assignments $B^{j_0}$ for some group $j_0$ given that the assignments for all other groups, denoted $B^{\backslash j_0}$, are held fixed. Let $m_i = \sum_{j,l} B_{il}^j$ denote number of local parameters assigned to the global parameter $i$, $m_i^{\backslash j_0} = \sum_{j \neq j_0, l} B_{il}^j$ be the same outside of group $j_0$ and let $L_{\backslash j_0}$ denote the number of unique global parameters corresponding to the

local parameters outside of $j_0$. The corresponding objective functions are given by

$$\mathcal{L}_{B^{\backslash j_0},\Theta}(B^{j_0}) = \log \mathbb{P}(B^{j_0} \mid B^{\backslash j_0})$$

$$- \sum_{i=1}^{L_{\backslash j_0}+L_{j_0}} \log \mathcal{H}\left(\tau + \sum_l B_{il}^{j_0} T(v_{j_0 l}) + \sum_{j\neq j_0,l} B_{il}^{j} T(v_{jl}), \sum_l B_{il}^{j} + m_i^{\backslash j_0} + n_0\right). \quad (8)$$

To arrive at a form of a linear sum assignment problem we define a *subtraction trick*:

**Proposition 1** (Subtraction trick). *When $\sum_l B_{il} \in \{0,1\}$ and $B_{il} \in \{0,1\}$ for $\forall i, l$, optimizing $\sum_i f(\sum_l B_{il} x_l + C)$ for $B$ is equivalent to optimizing $\sum_{i,l} B_{il}(f(x_l + C) - f(C))$ for any function $f$, $\{x_l\}$ and $C$ independent of $B$.*

*Proof.* This result simply follows by observing that both objectives are equal for any values of $B$ satisfying the constraint. $\qquad\square$

Applying the subtraction trick to (8) (conditions on $B$ are satisfied per (5)), we arrive at a linear sum assignment formulation $\mathcal{L}_{B^{\backslash j_0},\Theta}(B^{j_0}) = -\sum_{i,l} B_{il}^{j_0} C_{il}^{j_0}$, where the cost

$$C_{il}^{j_0} = \begin{cases} \log \frac{m_i^{\backslash j_0}}{\alpha+J-1-m_i^{\backslash j_0}} - \log \frac{\mathcal{H}\left(\tau+T(v_{j_0 l})+\sum_{j\neq j_0,l} B_{il}^{j} T(v_{jl}),1+m_i^{\backslash j_0}+n_0\right)}{\mathcal{H}\left(\tau+\sum_{j\neq j_0,l} B_{il}^{j} T(v_{jl}),m_i^{\backslash j_0}+n_0\right)}, & i \leq L_{\backslash j_0} \\ \log \frac{\alpha\gamma_0}{(\alpha+J-1)(i-L_{\backslash j_0})} - \log \frac{\mathcal{H}\left(\tau+T(v_{j_0 l}),1+n_0\right)}{\mathcal{H}(\tau,n_0)}, & L_{\backslash j_0} < i \leq L_{\backslash j_0} + L_j. \end{cases} \quad (9)$$

Terms on the left are due to $\log \mathbb{P}(B^{j_0} \mid B^{\backslash j_0})$. Details are provided in the supplement. Our algorithm consists of alternating the Hungarian algorithm with the above cost and hyperparameter optimization using $\log \mathbb{P}(B \mid v)$ from (6), ignoring $\mathbb{P}(B)$ as it is a constant with respect to hyperparameters. Specifically, the hyperparameter optimization step is

$$\hat{\tau}, \hat{n}_0 = \arg\max_{\tau,n_0} \sum_{i=1}^{L} \left( \log \mathcal{H}(\tau,n_0) - \log \mathcal{H}\left(\tau + \sum_{j,l} B_{il}^{j} T(v_{jl}), \sum_{j,l} B_{il}^{j} + n_0\right) \right), \quad (10)$$

where $B$ is held fixed. After obtaining estimates for $B$ and the hyperparameters $\Theta$, it only remains to compute global parameters estimates $\{\theta_i\}_{i=1}^{L} = \arg\max_{\theta_1,\dots,\theta_L} \mathbb{P}(\{\theta_i\}_{i=1}^{L}|B,v,\Theta)$. Given the assignments, expressions for hyperparameter and global parameter estimates can be obtained using gradient-based optimization. In Section 4.1 we give a concrete example where derivations may be done in closed form. Our method, Statistical Parameter Aggregation via Heterogeneous Matching (SPAHM, pronounced "spam"), is summarized as Algorithm 1.

---

**Algorithm 1** Statistical Parameter Aggregation via Heterogeneous Matching (SPAHM)

---

**input** Observed local $v_{jl}$, iterations number $M$, initial hyperparameter guesses $\hat{\tau}, \hat{n}_0$.
1: **while** not converged **do**
2:    **for** $M$ iterations **do**
3:       $j \sim \text{Unif}(\{1,\dots,J\})$.
4:       Form matching cost matrix $C^j$ using eq. (9).
5:       Use Hungarian algorithm to optimize assignments $B^j$, holding all other assignments fixed.
6:    **end for**
7:    Given $B$, optimize (10) to update hyperparameters $\hat{\tau}, \hat{n}_0$.
8: **end while**
**output** Matching assignments $B$, global atom estimates $\theta_i$.

---

## 4.1 Meta Models with Gaussian Base Measure

We present an example of how a statistical modeler may apply SPAHM in practice. The only choice modeler has to make is the prior over parameters of their local models, i.e. (4). In many practical scenarios (as we will demonstrate in the experiments section) model parameters are real-valued and the Gaussian distribution is a reasonable choice for the prior on the parameters. The Gaussian case is

further of interest as it introduces additional parameters. For simplicity we consider the 1-dimensional Gaussian, which is also straightforward to generalize to multi-dimensional isotropic case.

The modeler starts by writing the density

$$p_v(v \mid \theta, \sigma^2) = \frac{1}{\sqrt{2\pi\sigma^2}} \exp\left(-\frac{(v-\theta)^2}{2\sigma^2}\right) \text{ and noticing that}$$

$$h_\sigma(v) = \frac{1}{\sqrt{2\pi\sigma^2}} \exp\left(-\frac{v^2}{2\sigma^2}\right), \quad T_\sigma(v) = \frac{v}{\sigma^2}, \quad \mathcal{A}_\sigma(v) = \frac{\theta^2}{2\sigma^2}.$$

Here the subscript $\sigma$ indicates dependence on the additional parameter, i.e. variance. Next,

$$H_\sigma(\tau, n_0) = \left(\int \exp\left(\tau\theta - \frac{n_0\theta^2}{2\sigma^2}\right) d\theta\right)^{-1} = \left(\exp\left(\frac{\tau^2\sigma^2}{2n_0}\right)\sqrt{\frac{2\pi\sigma^2}{n_0}}\right)^{-1}$$

to ensure $p_\theta(\theta|\tau, n_0)$ integrates to unity. Hence,

$$\log H_\sigma(\tau, n_0) = -\frac{\tau^2\sigma^2}{2n_0} + \frac{\log n_0 - \log \sigma^2 - \log 2\pi}{2}.$$

These are all we need to customize (9) and (10) to the Gaussian case, which then allows the modeler to use Algorithm 1 to compute the shared parameters across the datasets. Note that in this case $\sum_{j,l} \log h_\sigma(v_{jl})$ should be added to eq. (10) if it is desired to learn the additional parameter $\sigma^2$. We recognize that not every exponential family allows for closed form evaluation of the prior normalizing constant $H_\sigma(\tau, n_0)$, however it remains possible to use SPAHM by employing Monte Carlo techniques for estimating entries of the cost (9) and auto-differentiation [2] to optimize hyperparameters.

Continuing our Gaussian example, we note that setting $\tau = \mu_0/\sigma_0^2$ and $n_0 = \sigma^2/\sigma_0^2$ we recover (1) as a density of a Gaussian random variable with mean $\mu_0$ and variance $\sigma_0^2$, as expected. The further benefit of the Gaussian choice is the closed-form solution to the hyperparameters estimation problem.

Under the mild assumption that $\sigma_0^2 + \sigma^2/m_i \approx \sigma_0^2 \ \forall i$ (i.e., global parameters are sufficiently distant from each other in comparison to the noise in the local parameters) we obtain

$$\hat{\mu}_0 = \frac{1}{L}\sum_{i=1}^{L}\frac{1}{m_i}\sum_{j,l} B_{il}^j v_{jl}, \quad \hat{\sigma}^2 = \frac{1}{N-L}\sum_{i=1}^{L}\left(\sum_{j,l} B_{il}^j v_{jl}^2 - \frac{(\sum_{j,l} B_{il}^j v_{jl})^2}{m_i}\right),$$

$$\hat{\sigma}_0^2 = \frac{1}{L}\sum_{i=1}^{L}\left(\frac{\sum_{j,l} B_{il}^j v_{jl}}{m_i} - \hat{\mu}_0\right)^2 - \sum_{i=1}^{L}\frac{\hat{\sigma}^2}{m_i},$$

where $N = \sum_j L_j$ is the total number of observed local parameters. The result may be verified by setting corresponding derivatives of eq. (10) $+ \sum_{j,l} \log h_\sigma(v_{jl})$ to 0 and solving the system of equations. Derivations are long but straightforward. Given assignments $B$, our example reduces to a hierarchical Gaussian model – see Section 5.4 of Gelman *et al.* [16] for analogous hyperparameter derivations. Finally we obtain

$$\theta_i = \frac{\mu_0\sigma^2 + \sigma_0^2 \sum_{j,l} B_{il}^j v_{jl}}{\sigma^2 + m_i\sigma_0^2}.$$

For completeness we provide cost expression corresponding to eq. (9): $C_{il}^{j_0} =$

$$\begin{cases} 2\log\frac{m_i^{\setminus j_0}}{\alpha+J-1-m_i^{\setminus j_0}} + \log\frac{m_i^{\setminus j_0}+\frac{\sigma^2}{\sigma_0^2}}{1+m_i^{\setminus j_0}+\frac{\sigma^2}{\sigma_0^2}} + \frac{\left(\frac{\mu_0}{\sigma_0^2}+\frac{v_{j_0l}}{\sigma^2}+\sum_{j\neq j_0,l} B_{il}^j\frac{v_{jl}}{\sigma^2}\right)^2\sigma^2}{1+m_i^{\setminus j_0}+\frac{\sigma^2}{\sigma_0^2}} - \frac{\left(\frac{\mu_0}{\sigma_0^2}+\sum_{j\neq j_0,l} B_{il}^j\frac{v_{jl}}{\sigma^2}\right)^2\sigma^2}{m_i^{\setminus j_0}+\frac{\sigma^2}{\sigma_0^2}}, \\ 2\log\frac{\alpha\gamma_0}{(\alpha+J-1)(i-L_{\setminus j_0})} + \log\frac{\sigma^2}{\sigma_0^2+\sigma^2} + \frac{(\mu_0/\sigma_0^2+v_{j_0l}/\sigma^2)^2\sigma^2}{1+\sigma^2/\sigma_0^2} - \frac{\mu_0^2}{\sigma_0^2}, \end{cases} \tag{11}$$

where first case is for $i \leq L_{\setminus j_0}$ and second is for $L_{\setminus j_0} < i \leq L_{\setminus j_0} + L_j$.

## 4.2 Convergence Analysis

**Lemma 1** (Algorithmic convergence). Algorithm 1 creates a sequence of iterates for which $\log \mathbb{P}(B \mid v)$ converges as the number of iterations $n \to \infty$. See Section B of the supplement for a proof sketch.

**Hyperparameter Consistency.**    While the exponential family hyperparameter objective function (10) is too general to be tractable, the consistency of the hyperparameter estimates can be analyzed for specific choices of distributional families. Following the specialization to Gaussian distributions in Section 4.1, the following result establishes that the closed-form hyperparameter estimates are consistent in the case of Gaussian priors, subject to the assignments $B$ being correct.

**Theorem 1.**  Assume that the binary assignment variables $B$ are known or estimated correctly. The estimator for $\hat{\mu}_0$ for the hyperparameter $\mu_0$ in the Gaussian case is then consistent as the number of global atoms $L \to \infty$. Furthermore, the estimators $\hat{\sigma}_0^2$ and $\hat{\sigma}^2$ for the hyperparameters $\sigma_0^2$ and $\sigma^2$ are consistent as the total number of global atoms with multiple assignments $\sum_{i=1}^{L} I((\sum_{j,l} B_{il}^j) > 1) \to \infty$, where $I(\cdot)$ is the indicator function. See Section C of the supplement for a detailed proof.

# 5   Experiments

**Simulated Data.**   We begin with a correctness verification of our inference procedure via a simulated experiment. We randomly sample $L = 50$ global centroids $\theta_i \in \mathbb{R}^{50}$ from a Gaussian distribution $\theta_i \sim \mathcal{N}(\mu_0, \sigma_0^2 \mathbf{I})$. We then simulate $j = 1, \ldots, J$ heterogeneous datasets by picking a random subset of global centroids and adding white noise with variance $\sigma^2$ to obtain the "true" local centroids, $\{v_{jl}\}_{l=1}^{L_j}$ (following generative process in Section 3 with Gaussian densities). Then each dataset is sampled from a Gaussian mixture model with the corresponding set of centroids. We want to estimate global centroids and parameters $\mu_0, \sigma_0^2, \sigma^2$. We consider two basic baselines: k-means clustering of all datasets pooled into one (k-means pooled) and k-means clustering of local centroid estimates (this can be seen as another form of parameter aggregation - i.e. k-means "matching"). Both, unlike SPAHM, enjoy access to true $L$.

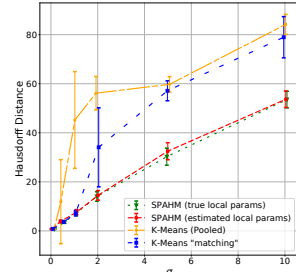

Figure 1: Increasing noise $\sigma$

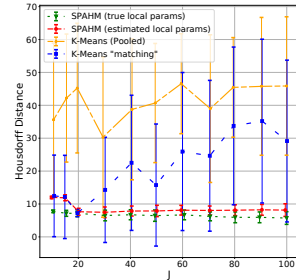

Figure 2: Increasing $J$

To obtain local centroid estimates for SPAHM and k-means "matching", we run (another) k-means on each of the simulated datasets. Additionally to quantify how local estimation error may effect our approach, we compare to SPAHM using true data generating local centroids. To measure the quality of different approaches we evaluate Hausdorff distance between the estimates and true data generating *global* centroids. This experiment is presented in Figures 1 and 2. White noise variance $\sigma$ implies degree of heterogeneity across $J = 20$ datasets and as it grows the estimation problem becomes harder, however SPAHM degrades more gracefully than baselines. Fixing $\sigma^2 = 1$ and varying number of datasets $J$ may make the problem harder as there is more overlap among the datasets, however SPAHM is able to maintain low estimation error. We empirically verify hyperparameter estimation quality in the supplement.

**Gaussian Topic Models.**   We present a practical scenario where problem similar to our simulated experiment arises — learning Gaussian topic models [7] where local topic models are learned from the Gutenberg dataset comprising 40 books. We then build the global topic model using SPAHM. We use basic k-means with $k = 25$ to cluster word embeddings of words present in a book to obtain local topics and then apply SPAHM resulting in 155 topics. We compare to the Gibbs sampler of Das *et al.* [7] in terms of the UCI coherence score [27], $-2.1$ for SPAHM and $-4.6$ for [7], where higher is better. Besides, [7] took 16 hours to run 100 MCMC iterations while SPAHM + k-means takes only 40 seconds, ***over* 1400 *times faster***. We present topic interpretation in Fig. 3 and defer additional details to the Supplement.

**Gaussian Processes.**   We next demonstrate the effectiveness of our approach on the task of temperature prediction using Sparse Gaussian Processes (SGPs) [1]. For this task, we utilize the GSOD data available from the National Oceanic and Atmospheric Administration[2] containing the daily global surface weather summary from over 9000 stations across the world. We limit the geography of our data to the United States alone and also filter the observations to the year 2015 and after. We further

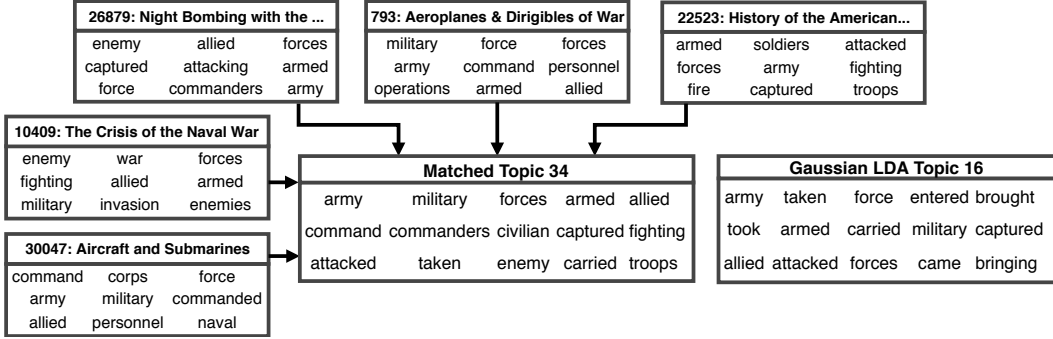

Figure 3: Topic related to war found by SPAHM and Gaussian LDA. The five boxes pointing to the Matched topic represent local topics that SPAHM fused into the global one. The headers of these five boxes state the book names along with their Gutenberg IDs.

Table 1: Temperature prediction using sparse Gaussian Processes

| EXPERIMENT SETUP | RMSE (SELF) | RMSE (ACROSS USA) |
|---|---|---|
| GROUP SGPS WITH 50 LOCAL PARAMETERS | $5.509 \pm 0.0135$ | $14.565 \pm 0.0528$ |
| SPAHM WITH $289 \pm 9.8$ GLOBAL PARAMETERS | $5.860 \pm 0.0390$ | $8.917 \pm 0.1988$ |
| GROUP SGPS WITH 300 LOCAL PARAMETERS | $5.267 \pm 0.0084$ | $15.848 \pm 0.0303$ |

select the following 7 features to create the final dataset - date (day, month, year), latitude, longitude, and elevation of the weather stations, and the previous day's temperature. We consider states as datasets of wheather stations observations.

We proceed by training SGPs on each of the 50 states data and evaluate it on the test set consisting of a random subset drawn from all states. Such locally trained SGPs do not generalize well beyond their own region, however we can apply SPAHM to match local inducing points along with their response values and pass it back to each of the states. Using inducing points found by SPAHM, local GPs gain ability to generalize across the continent while maintaining comparable fit on its own test data (i.e. test data sampled only from a corresponding state). We summarize the results across 10 experiment repetitions in Table 1. In addition, we note that Bauer *et al.* [1] previously showed that increasing number of inducing inputs tends to improve performance. To ensure this is not the reason for strong performance of SPAHM we also compare to local SGPs trained with 300 inducing points each.

**Hidden Markov Models.** Next, we consider the problem of discovering common structure in collections of related MoCAP sequences collected from the CMU MoCap database (`http://mocap.cs.cmu.edu`). We used a curated subset [14] of the data from two different subjects each providing three sequences. This subset comes with human annotated labels which allow quantitative comparisons. We performed our experiments on this annotated subset. For each subject, we trained an independent 'sticky' HDP-HMM [13] with Normal Wishart likelihoods. We used memoized variational inference [24] with random restarts and merge moves to alleviate local optima issues (see supplement for details about parameter settings and data pre-processing). The trained models discovered nine states for the first subject and thirteen for the second. We then used SPAHM to match local HDP-HMM states across subjects and recovered fourteen global states. The matching was done on the basis of the posterior means of the local states.

The matched states are visualized in Figure 4 (right) and additional visualizations are available in the supplement. We find that SPAHM correctly recognizes similar activities across subjects. It also creates singleton states when there are no good matches. For instance, "up-downs", an activity characterized by distinct motion patterns is only performed by the second subject. We correctly do not match it with any of the activities performed by the first subject. The figure also illustrates a limitation of our procedure wherein poor local states can lead to erroneous matches. Global states five and six are a combination of "toe-touches" and "twists". State five combines exaggerated motions to the right while state six is a combination of states with motions to the left. Although the toe-touch

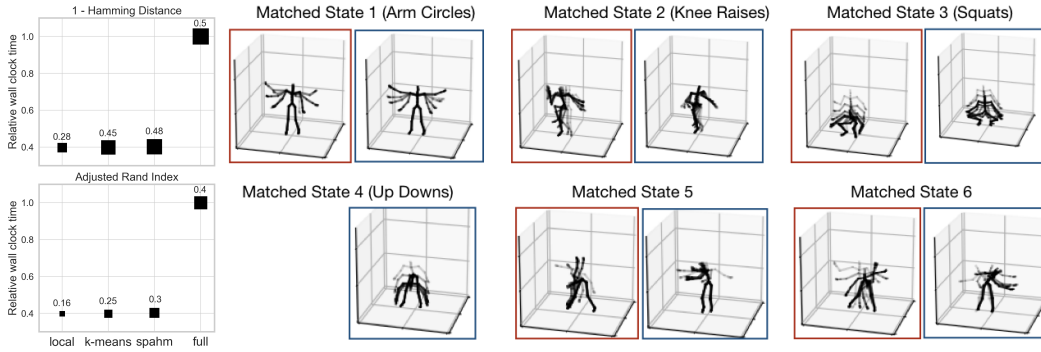

Figure 4: **BBP discovers coherent global structure from MoCap Sequences**. We analyze three MoCAP sequences each from two subjects performing a series of exercises. Some exercises are shared between subjects while others are not. Two HDP-HMMs were fit to explain the sequences belonging to each subject independently. *Left*: We show the fraction of Full HDP-HMM wall clock time taken by various competing algorithms. The area of each square is proportional to 1 - normalized hamming distance and adjusted rand index in the top and bottom plots. The actual values are listed above each square. Larger squares indicate closer matches to ground truth. At less than half the compute SPAHM produces similar segmentations to the full HDP-HMM while improving significantly on the local models and k-means based matching. *Right*: Typical motions associated with the matched states from the two models are visualized in the red and blue boxes. Skeletons are visualized from contiguous segments of at least 0.5 seconds of data as segmented by the MAP trajectories.

activities exhibit similar motions, the local HDP-HMM splits them into different local states. Our matching procedure only allows local states to be matched across subjects and not within. As a result, they get matched to oversegmented "twists" with similar motions.

We also quantified the quality of the matched solutions using normalized hamming distance [14] and adjusted rand index [28]. We compare against local HDP-HMMs as well as two strong competitors, an identical sticky HDP-HMM model but trained on all six sequences jointly, and an alternate matching scheme based on $k$-means clustering that clusters together states discovered by the individual HDP-HMMs. For $k$-means, we set $k$ to the ground truth number of activities, twelve. The results are shown in Figure 4 (left). Quantitatively results show that SPAHM does nearly as well as the full HDP-HMM at less than half the amount of compute time. Note that SPAHM may be applied to larger amount of sequences, while full HDP-HMM is limited to small data sizes. We also outperform the k-means scheme despite cheating in its favor by providing it with the true number of labels.

## 6 Conclusion

This work presents a statistical model aggregation framework for combining heterogeneous local models of varying complexity trained on federated, private data sources. Our proposed framework is largely model-agnostic requiring only an appropriately chosen base measure, and our construction assumes only that the base measure belongs to the exponential family. As a result, our work can be applied to a wide range of practical domains with minimum adaptation. A possible interesting direction for future work will be to consider situations where local parameters are learned across datasets with a time stamp in addition to the grouping structure.

## Footnotes

[1]Code: `https://github.com/IBM/SPAHM`

[2]https://data.noaa.gov/dataset/dataset/global-surface-summary-of-the-day-gsod

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
