[Supplementary Material]

# Supplement for Statistical Model Aggregation via Parameter Matching

**Mikhail Yurochkin**[1,2]
mikhail.yurochkin@ibm.com

**Mayank Agarwal**[1,2]
mayank.agarwal@ibm.com

**Soumya Ghosh**[1,2,3]
ghoshso@us.ibm.com

**Kristjan Greenewald**[1,2]
kristjan.h.greenewald@ibm.com

**Trong Nghia Hoang**[1,2]
nghiaht@ibm.com

IBM Research,[1] MIT-IBM Watson AI Lab,[2] Center for Computational Health[3].

## A  IBP prior term derivation

We provide derivations for $\log \mathbb{P}(B^{j_0} \mid B^{\backslash j_0})$ for the cost expression in Eq. (9) of the main text.

First note that due to exchangeability of the IBP, customer $\backslash j_0$ may be considered as the last one, hence:

$$
\begin{aligned}
\log \mathbb{P}(B^{j_0} \mid B^{\backslash j_0}) = \\
\sum_{i=1}^{L_{\backslash j_0}} \left( \left( \sum_{l=1}^{L_{j_0}} B_{il}^{j_0} \right) \log \frac{m_i^{\backslash j_0}}{\alpha + J - 1} + \left( 1 - \sum_{l=1}^{L_{j_0}} B_{il}^{j_0} \right) \log \frac{\alpha + J - 1 - m_i^{\backslash j_0}}{\alpha + J - 1} \right) + \\
\sum_{i=L_{\backslash j_0}+1}^{L_{\backslash j_0}+L_{j_0}} \sum_{l=1}^{L_{j_0}} B_{il}^{j_0} \left( \log \frac{\alpha \gamma_0}{\alpha + J - 1} - \log(i - L_{\backslash j_0}) \right).
\end{aligned} \tag{1}
$$

It is now easy to see that when $i \le L_{\backslash j_0}$, the contribution of the IBP prior is $\log \frac{m_i^{\backslash j_0}}{\alpha + J - 1 - m_i^{\backslash j_0}}$, and when $L_{\backslash j_0} < i \le L_{\backslash j_0} + L_{j_0}$, it is $\log \frac{\alpha \gamma_0}{(\alpha + J - 1)(i - L_{\backslash j_0})}$.

## B  Proof of Lemma 1

**Lemma 1** (Algorithmic convergence). Algorithm 1 (of the main text) creates a sequence of iterates for which $\log \mathbb{P}(B \mid v)$ converges as the number of iterations $n \to \infty$.

*Proof.* Since the Hungarian algorithm finds a globally optimal solution to the assignment problem for $B^j$, each Hungarian step must yield a larger or equal objective function value ($\log \mathbb{P}(B \mid v)$) compared to the previous $B^j$ (where $B^{\backslash j}$ and the hyperparameters are held fixed). Similarly, the hyperparameter optimization step is assumed to find a global optimum (for Gaussian priors it is closed form) of $\log \mathbb{P}(B \mid v)$ with $B$ fixed and must therefore not decrease the objective $\log \mathbb{P}(B \mid v)$. Therefore, each step in Algorithm 1 cannot decrease $\log \mathbb{P}(B \mid v)$. Since the $\log \mathbb{P}(B \mid v)$ is bounded from above by 0 (since $\mathbb{P}(B \mid v)$ is discrete-valued), this implies that Algorithm 1 creates a sequence of iterates for which the objective function value converges. □

## C   Proof of Theorem 1: Gaussian hyperparameter consistency

First consider $\hat{\mu}_0$. Recall that

$$\hat{\mu}_0 = \frac{1}{L} \sum_{i=1}^{L} \frac{1}{m_i} \sum_{j,l} B_{il}^j v_{jl}.$$

Hence, $\mathbb{E}\hat{\mu}_0 = \mu_0$ since the marginal expectation $\mathbb{E}v_{jl} = \mu_0$ and the $B_{il}^j$ are assumed known. Since the $v_{jl}$ are Gaussian with bounded variance, and the underlying $L$ global atoms are independent, $\hat{\mu}_0$ will concentrate around its mean when $L \to \infty$. Hence $\hat{\mu}_0$ is consistent as desired.

Next, consider $\hat{\sigma}^2$. Recall that

$$\hat{\sigma}^2 = \frac{1}{N - L} \sum_{i=1}^{L} \left( \sum_{j,l} B_{il}^j v_{jl}^2 - \frac{(\sum_{j,l} B_{il}^j v_{jl})^2}{m_i} \right).$$

Since the $v_{jl}$ are Gaussian with bounded variance and at least $L$ of them are independent, and the $B_{il}^j$ are binary, $\hat{\sigma}^2$ concentrates around its expectation as the total number of global atoms with multiple assignments $\sum_{i=1}^{L} I((\sum_{j,l} B_{il}^j) > 1) \to \infty$ where $I(\cdot)$ is the indicator function. This follows by the Bernstein inequality for subexponential random variables [6]. Now

$$\mathbb{E}\hat{\sigma}^2 = \frac{\sum_{i,j,l} B_{il}^j \mathbb{E}v_{jl}^2 - \sum_{i=1}^{L} \mathbb{E}\frac{(\sum_{j,l} B_{il}^j v_{jl})^2}{m_i}}{N - L}$$

$$= \frac{\sum_{i,j,l} B_{il}^j (\mu_0^2 + \sigma_0^2 + \sigma^2) - \sum_{i=1}^{L} \left[ (\mu_0^2 + \sigma_0^2)m_i + \sigma^2 \right]}{N - L}$$

$$= \frac{\sum_{i,j,l} B_{il}^j \sigma^2 - \sum_{i=1}^{L} \sigma^2}{N - L}$$

$$= \sigma^2.$$

Recalling that $N$ is the total number of local parameters, hence $\sum_{i,j,l} B_{il}^j = N$ and it follows that given the $B_{il}^j$, $\hat{\sigma}^2$ is consistent.

Finally, we consider the $\hat{\sigma}_0^2$ estimate, which depends on $\hat{\mu}_0$ and $\hat{\sigma}^2$. Recall that

$$\hat{\sigma}_0^2 = \frac{1}{L} \sum_{i=1}^{L} \left( \left( \frac{\sum_{j,l} B_{il}^j v_{jl}}{m_i} - \hat{\mu}_0 \right)^2 - \frac{\hat{\sigma}^2}{m_i} \right).$$

Note that since the $v_{jl}$ are Gaussian, for fixed $B_{il}^j$, $\hat{\sigma}_0^2$ will concentrate around its expectation if $L \to \infty$ (again by the Bernstein inequality for subexponential random variables [6]). Note further that if $\hat{\mu}_0 = \mu_0$ and $\hat{\sigma} = \sigma$, $\mathbb{E}\hat{\sigma}_0^2 = \sigma_0^2$ since the variance

$$\text{Var}\left[ \frac{\sum_{j,l} B_{il}^j v_{jl}}{m_i} \right] = \sigma_0^2 + \frac{\sigma^2}{m_i},$$

which holds since the $B_{il}^j$ are binary. Hence by smoothness, if $\hat{\mu}_0$ and $\hat{\sigma}^2$ are consistent, $\hat{\sigma}_0^2$ will be as well.   $\square$

## D   Hyperparameter estimation quality

Using our simulated experiments we verify the correctness of our hyperparameter estimation procedure and statement of Theorem 1. In Figure 1 we vary $\sigma$ and measure relative estimation error, which is defined as absolute error normalized by the true value. In this experiment we utilized k-means estimates of the local centroids for SPAHM. We see that when the variance of local centroids with respect to the global ones is not too big, SPAHM produces high quality estimates as well as precision in recovering true number of global parameters. It is interesting to note the almost exact $\sigma$ recovery —

Figure 1: Verification of the hyperparamter estimation quality

this is because estimate for $\sigma$ presented in Section 4.1 of the main text did not require the assumption that $\sigma_0^2 + \sigma^2/m_i \approx \sigma_0^2$ $\forall i$ and was derived exactly. For other hyperparameters the assumption was needed and appears to introduce minor bias. Additionally we note that hyperparameters $\alpha$ and $\gamma_0$ need to be set by the modeler as they represent prior beliefs regarding the amount of sharing of global parameters (i.e. $\alpha$) across datasets and their quantity (i.e. $\gamma_0$). In all our experiments we set $\alpha = 1$ and $\gamma_0 = 1$, except sparse Gaussian process experiment where we set $\gamma_0 = 50$ as we expected larger number of inducing points needed to model weather across all 50 states.

## E    HDP-HMM details

Our HMM models use multivariate Normal-Wishart observation models and Hierarchical Dirichlet process allocation models. The state specific transition probabilities $\pi_k$ are drawn according to the following generative process. First, we draw $\beta \sim \text{GEM}(\gamma)$ from the stick breaking distribution. That is,

$$\beta_j = \nu_j \prod_{l=1}^{j-1}(1 - \nu_l); \quad \nu_j \mid \gamma \sim \text{Beta}(1, \gamma); \quad j = 1, 2, \ldots, \tag{2}$$

We then draw $\pi_k$ from a Dirichlet process with a discrete base measure shared across states,

$$\pi_k \mid \eta, \kappa, \beta \sim \text{DP}(\eta + \kappa, \frac{\eta\beta + \kappa\delta_k}{\eta + \kappa}); \quad k = 1, 2, \ldots, \tag{3}$$

where $\eta$ is a concentration parameter and $\kappa$ is a "stickyness" parameter which encourages state persistence. The latent states for a particular sequence then evolve as $z_t$, evolve as $z_{t+1} \mid z_t, \{\pi_k\}_{k=1}^{\infty} \sim \pi_{z_t}$. Finally, observations at time step $t$, $y_t \in \mathbb{R}^D$ are drawn from a Normal Wishart distribution,

$$
\begin{aligned}
\mu_k \mid \mu_0, \lambda, \Lambda_k &\sim \mathcal{N}(\mu_0, (\lambda\Lambda_k)^{-1}) \\
\Lambda_k \mid S, n_0 &\sim \text{Wishart}(n_0, S) \\
y_t \mid z_t = k &\sim \mathcal{N}(y_t \mid \mu_k, \Lambda_k^{-1})
\end{aligned}
\tag{4}
$$

For all our experiments, we set $\kappa$ to 10.0, $\gamma = 5$. and $\eta = 0.5$. For the observation model, we set $n_0 = 1$ and $S$ to an identity matrix $\mathbf{I}$, encoding our belief that $\mathbb{E}[\Lambda_k^{-1}] = \mathbf{I}$.

### E.1    MoCAP data details

We consider the problem of discovering common structure in collections of related time series. Although such problems arise in a wide variety of domains, here we restrict our attention to data captured from motion capture sensors on joints of people performing exercise routines. We collected

this data from the CMU MoCap database (`http://mocap.cs.cmu.edu`). Each motion capture sequence in this database consists of 64 measurements of human subjects performing various exercises. Following [2], we select 12 measurements deemed most informative for capturing gross motor behaviors: body torso position, neck angle, two waist angles, and a symmetric pair of right and left angles at each subjects shoulders, wrists, knees, and feet. Each MoCAP sequence thus provides a 12-dimensional time series. We use a curated subset [2] of the data from two different subjects each providing three sequences. In addition to having several exercise types in common this subset comes with human annotated labels allowing for easy quantitative comparisons across different models.

### E.2   Metrics

**Normalized Hamming distance**   We follow [2] and compute the normalized Hamming distance between the MAP segmentation and the human-provided ground truth annotation using the optimal alignment of each ground truth state to a predicted state. Normalized Hamming distance then measures the fraction of time steps where the labels of the ground-truth and estimated segmentations disagree.

**Adjusted Rand Index**   Rand index [5] is commonly used to measure the quality of a partition with respect to a ground truth partitioning. It is defined as,

$$R = \frac{a + b}{\binom{n}{2}}, \tag{5}$$

where a is the number of pairs of elements that are in the same subset in the two partitions and b is the number of pairs of elements that are in different subsets in the two partitions. The denominator is the total number of pairs. The adjusted rand index is computed as,

$$ARI = \frac{R - \mathbb{E}(R)}{\max(R) - \mathbb{E}(R)} \tag{6}$$

### E.3   Additional Results

In Figure 2, we present additional matched states discovered by SPAHM.

## F   Topic Modeling Experiments

We also evaluate SPAHM on the task of topic modeling. Here, we randomly select 40 books from Project Gutenberg [1], primarily in 2 unrelated domains to introduce heterogeneity in the data - World War I and Astronomy. For modeling using SPAHM, we first extract 25 topics from each book using k-means (independently for each book), and then match the topics extracted from each book to produce the global topics for the entire corpus. As a baseline measure, we compare our method to Gaussian LDA [1] trained on the whole corpus of 40 books. Unlike SPAHM, Gaussian LDA requires the number of global topics to be specified a priori. To keep the evaluations fair, we train the Gaussian LDA for 100 MCMC iterations to extract 150 topics which is similar to the number of topics extracted by SPAHM (155 topics).

We quantitatively compare the two approaches by calculating the UCI coherence [4] scores over the Gutenberg dataset [3] consisting of 3000 books separate from the ones used to train the two models. The coherence score along with the time taken by each model is presented in Table 1, where higher number implies more coherent topics. As a qualitative measure of the topics found by the two methods, we select a common topic related to "war" found by both SPAHM and Gaussian LDA, and look at the closest 15 words to the topic found by each method. We observed that while the general theme of the topic is captured by both models, Gaussian LDA topic consists of uninformative words such as "taken", "brought", "took", among the informative words like "army", "military", "command". On the other hand, 14 of the top 15 words extracted by SPAHM are very relevant to the topic of "war". We present the corresponding topics in Figure 3 (same as in the main text).

Figure 2: Additional matched states discovered by SPAHM

Table 1: Coherence scores and runtimes for estimating Gutenberg topics

|  | SPAHM | Gaussian LDA |
|---|---|---|
| UCI Coherence score | -2.0967 | -4.5956 |
| Runtime | 42 sec | $\sim 600$ sec/iteration |

Figure 3: Topic related to war found by SPAHM and Gaussian LDA. The five boxes pointing to the Matched topic represent local topics that SPAHM fused into the global one. The headers of these five boxes state the book names along with their Gutenberg IDs.

## Footnotes

[1]`https://www.gutenberg.org`