[Reviews · NeurIPS 2019]

Reviewer 1



The choice of the exponential family is the same for all the local parameters, so I am assuming the method is suited to independent partitions of the data? When I first started reading the paper, I was expecting some sort of aggregation of different methods on the same data and in that case, we may be dealing with completely different parameters and different distributions. Possibility of such extensions should be discussed. I think the idea of sharing global parameters in Q_j is quite restrictive. For example if Q_js are moderately high dimensional, it may be desirable to only share only a few coordinates while keeping the other coordinates distinct, much in the way of the Local partition processes: https://academic.oup.com/biomet/article-abstract/96/2/249/249850?redirectedFrom=fulltext In the simulation study, the authors may also consider comparing with Chinese Restaurant process based clustering which does not require the true value of L.

Reviewer 2



I think the idea is interesting and the motivation for this problem is important. However, there's other meta-model Bayesian methods that I think the authors should consider discussing in their paper: 1.) Bayesian inference in hierarchical models by combining independent posteriors (Dutta, Blomstedt, Kaski, 2016) 2.) Meta-analysis of Bayesian analyses (Blomstedt et al., 2019) 3.) Differentially Private Bayesian Learning on Distributed Data (Heikkila et al. 2017) Also, perhaps a useful baseline comparison to your method would be a simple one level hierarchical extension of sharing features, which should be possible as the features are assumed Gaussian in each of the experiments, and compare the results against your method. This could better illustrate why your method is necessary in terms of computation time, privacy preserving, etc. I think it's quite a nice idea considering it's designed for structures like mixture models which I think is quite novel for this type of work.

Reviewer 3



Originality: The beta-Bernoulli interpretation for model aggregation is new to me. To briefly summarize, the paper uses a subset of global model parameters for local groups, and the subset selection process is modeled as beta-Bernoulli with a matching afterward. This method is novel to me, but I still have a few questions. 1) The model requires performing alternative updates for each group, is it possible to parallelize the algorithm? 2) It would be non-trivial to adaptively tune the cardinality of C_j. How do you do that in your experiments? If |C_j| diverges among groups, can your method accurately estimate |C_j| for each data group j? Overall I think the paper totally makes sense to me, but several details need to be verified. Quality: This paper is well-written, and I have not found any technical error within the paper. Significance: I think all experiments in this paper make sense to me but not quite enough since the paper does not compare over their method with provably convergent methods such as online learning methods (e.g. stochastic variational inference). So it is hard for me to judge the total quality of this novel learning strategy. Clarity: This paper is well-written and clearly explained.

[Author Response · NeurIPS 2019]

We thank the reviewers for their time, their valuable and encouraging feedback, and their recommendations for improvement. We remain confident that our work is of strong interest to the NeurIPS community and easily can incorporate the suggested changes in a revision for the conference. Answers to specific comments appear below.

**Related literature.** We thank **R1** and **R2** for pointing at several related papers. We believe that our model and inference techniques are substantially different, however we agree that all of the mentioned papers are relevant. We present a brief discussion below and we will add an extended discussion (and citations) to our paper.

**R1** suggested that Local Partition Process (LPP) of Dunson (2009) allows for sharing a subset of coordinates, which may be beneficial. We note that LPP is applied in a regression-like problem where there is a *single* global parameter - a vector of regression coefficients, and each dataset selects a (sparse) subset of the coordinates of this vector. In our work there is a *collection* of global parameters and each dataset selects an (also sparse) subset of these global parameters via the Bernoulli process, i.e. $Q_j$. On a high level, both models perform a sparse subset selection, however there are significant differences in modeling goals and inference. We suspect it might be possible to apply our model in the problem studied by Dunson and apply LPP in our setting, however it remains to be seen whether inference with LPPs can be generalized to the local models with inherent *permutation invariance* (mixtures, HMMs, etc.) that we consider. To clarify, in Dunson (2009), regression coefficients are naturally aligned across datasets as they are ordered according to the data coordinates; in our work, mixture components may be ordered arbitrarily for each dataset. What is perhaps a more worthy direction for future work is to develop a model capable of *both* selecting from a collection of global parameters and their coordinates.

**R2** mentioned a series of papers studying *meta-analysis of Bayesian analyses* applicable to random effects, linear regression, and other similar models. The key difference in our work is that we consider models with inherent *permutation invariant* structure of the parameter space: we demonstrate examples with mixtures, topic models, HMMs, and sparse GPs. Permutation invariance leads to inferential challenges associated with finding correspondences across sets of local parameters and learning the size of the global model, which are addressed in our work. On the other hand, it is not clear how the approach of Dutta et al. (2016) can be applied to models such as mixtures. The work of Heikkila et al. (2017) and Blomstedt et al. (2019) have similar modeling limitations, however they suggest interesting directions for future work: how to strengthen privacy preserving properties of SPAHM to guarantee *differential privacy* as in former, and how to generalize SPAHM to aggregate *local posteriors* instead of parameters as in the latter.

**Baselines.** **R3** asked for comparison to stochastic variational inference (SVI). In the paper we do compare against (memoized) online variational inference (see line 274) for the HDP-HMM models, which is the state-of-the-art for inference in such models and outperforms SVI. We also compare against a Gibbs sampler (line 245) for the Gaussian topic models. In both cases, SPAHM either outperforms or performs comparably while being significantly faster. Here, using SVI for inference, we compare to Chinese Restaurant process (CRP) in simulations as requested by **R1**. We also compare to CRP fitted with local centroid estimates, alike meta-modeling suggested by **R2**. This experiment is an extension of the Figure 1 of our paper. CRP performance is similar to k-means (which is expected as we have been fitting k-means with true $L$ in our experiments) and is inferior to SPAHM.

**R2**, if we understood correctly, suggested we compare to a full Bayesian hierarchical model and to other meta-modeling approaches to illustrate why our method is necessary. We believe that such results are contained in the paper. For example, in the Gaussian topic models experiment SPAHM is over 1400 times faster than hierarchical model inference with a Gibbs sampler (see lines 247-248). For the meta-modeling, we considered k-means clustering of the *local parameters* as a basic baseline: SPAHM outperforms this baseline in the motion capture experiment (see Fig. 4 left and lines 291-298) and in simulation studies (see k-means "matching" in Figures 1 and 2). In the figure presented in this rebuttal we also considered CRP "matching" as another meta-modeling baseline approach.

**Model and inference clarifications.** **R1** asked about the data partitions — we assume that local datasets (and corresponding parameters) are independently but *not* identically distributed. For example, our method can aggregate topics learned from datasets generated with *different* numbers of topics and even different topic models.

**R3** asked about learning the cardinality of $\mathcal{C}_j$ and parallelizing the algorithm. It is important to clarify that our approach performs *meta-modeling*. This means that first each dataset is processed independently and *in parallel* to obtain local sets of parameters. For a dataset $j$, there are $\mathrm{card}(\mathcal{C}_j)$ parameters, where $\mathrm{card}(\mathcal{C}_j)$ may be a hyperparameter or can be learned by applying an appropriate Bayesian nonparametric model *locally*, as we've done in our motion capture experiment. Then, these parameters serve as *input* to our algorithm and are not being updated. Our algorithm non-parametrically learns the global set of parameters and its size, allowing for data privacy and significant speedups compared to full Bayesian hierarchical learning (e.g., in the topic modeling experiment, our method is 1400 times faster than full hierarchical inference - see lines 247-248).

[Meta-Review · NeurIPS 2019]

The reviewers recommend accepting the paper. Their general impression is that the beta-Bernoulli process is being used in a unique way, in that each BeP selects local parameters for a group of data rather than the features present in a single observation. I agree with the reviewers that the paper can be accepted, since it's well-written and discusses a variety of applications (albeit at a very high level). However, the technical review and presentation of the model is elementary at best, and the proposed method is very simple, and it's often not obvious how it's being applied in the applications discussed. However, among borderline papers this one has the advantage of being understandable, well-written and mature in its approach to Bayesian ML.